# Chemical Constituents from *Streblus taxoides* Wood with Their Antibacterial and Antityrosinase Activities Plus in Silico Study

**DOI:** 10.3390/antibiotics12020319

**Published:** 2023-02-03

**Authors:** Kedsaraporn Parndaeng, Thanet Pitakbut, Chatchai Wattanapiromsakul, Jae Sung Hwang, Wandee Udomuksorn, Sukanya Dej-adisai

**Affiliations:** 1Department of Pharmacognosy and Pharmaceutical Botany, Faculty of Pharmaceutical Sciences, Prince of Songkla University, Hat-Yai 90112, Songkhla, Thailand; 2Pharmaceutical Biology, Department of Biology, Friedrich-Alexander-Universität Erlangen-Nürnberg (FAU), Staudtstr. 5, 91058 Erlangen, Germany; 3Department of Genetic Engineering & Graduate School of Biotechnology, College of Life Sciences, Kyung Hee University, Youngin 17104, Republic of Korea; 4Pharmacology Program, Division of Health and Applied Science, Faculty of Science, Prince of Songkla University, Hat-Yai 90112, Songkhla, Thailand

**Keywords:** Moraceae, *Streblus taxoides*, antityrosine activity, antibacterial activity, melanin content, intracellular tyrosinase activity

## Abstract

Hyperpigmentation frequently occurs after inflammation from bacterial infection. Thus, the inhibition activity of tyrosinase, the key enzyme to catalyze the melanogenesis and/or inhibition of bacterial infection, could decrease melanin production. Hence, the potential inhibitors could be discovered from natural products. ω-Hydroxymoracin C (**1**), a new compound with two other 2-arylbenzofurans, i.e., moracin M (**2**) and moracin C (**3**), and two stilbenes, i.e., 3, 4, 3’, 5′-tetrahydroxybibenzyl (**4**) and piceatannol (**5**), were isolated from the wood of *Streblus taxoides*. Compound **4** showed a strong inhibitory activity against tyrosinase enzyme with an IC_50_ value of 35.65 µg/mL, followed by compound **2** with an IC_50_ value of 47.34 µg/mL. Conversely, compound **1**, **3** and **5** showed moderate activity, with IC_50_ values of 109.64, 128.67 and 149.73 µg/mL, respectively. Moreover, compound **1** and **3** showed an antibacterial effect against some *Staphylococcus* spp. Thus, the isolated compounds exhibited potential antityrosine and antibacterial effects. Additionally, an in silico study was performed in order to predict theoretical molecular interactions between the obtained metabolites from *S. taxoides* and tyrosinase as an extended in vitro enzyme binding assay experiment.

## 1. Introduction

Gram-positive pathogenic bacteria are the main cause of human skin diseases; for example, *Staphylococcus aureus* and *S*. *epidermidis* can be a cause of impetigo, folliculitis and furunculosis [1] and *Cutibacterium acnes* can be a cause of skin inflammation, such as papules, pustules and so on [2]. Acne vulgaris or acne inflammation is one of the dermal skin infection diseases which causes negative social and psychological effects on sufferers. The facial skin infection caused by *Cutibacterium acnes*, *Malassezia furfur*, *S. epidermidis* and *S. aureus* can cause acne inflammation [3,4]. Interleukins (ILs) and some inflammatory mediators can stimulate tyrosinase activity and melanin synthesis by modulating proliferation and differentiation of human epidermal melanocytes and can also promote melanogenesis-related gene expression directly or indirectly [5,6,7,8,9]. Prostaglandin E2 (PGE2) and PGF2α are involved in all types of skin inflammation, which can stimulate melanocyte dendrite formation through a cAMP-dependent pathway, activate tyrosinase in melanocytes, increase melanin secretion, and induce pigmentation [10,11]. Conversely, IL-18 increases tyrosinase activity and upregulates tyrosinase-related protein 1 (TRP-1) and TRP-2 expression, IL-33 promotes microphthalmia-associated transcription factor (MITF), (tyrosinase) TYR, TRP-1 and TRP-2 expression by activating the p38/mitogen-activated protein kinase (MAPK) and protein kinase A (PKA) pathways and IL-1α combines keratinocyte growth factor (KGF) to increase melanin deposition [10]. Moreover, histamine induces morphologic changes and melanin synthesis of human melanocytes by PKA activation via H₂ receptor-mediated cAMP accumulation [12].

Melanin is the main pigment in mammals, found in skin, hair and eyes [13,14]. The major function of melanin is to provide protection against ultraviolet (UV) radiation, but abnormal melanin (hyperpigmentation) can lead to skin disorders [14]. Melanin is synthesized through a complex pathway, i.e., melanogenesis in melanosomes [15]. The regulation of melanogenesis is controlled by a variety of paracrine cytokines, including α-melanocyte-stimulating hormone (α-MSH), stem cell factor (SCF), endothelin-1 (ET-1), nitric oxide (NO), adrenocorticotropic hormone (ACTH), prostaglandins, thymidine dinucleotide and histamine. All factors induce melanogenesis through diverse signaling pathways by activating the expression and activation of pigment-related proteins such as microphthalmia-associated transcription factor (MITF); the master regulator of melanogenesis in melanocytes via binding to the M box of a promoter region and regulating the gene expression of tyrosinase (TYR), tyrosine-related protein-1 (TRP-1) and tyrosine-related protein-2 (TRP-2) [8,9,10]. There are three enzymes involved in the melanogenesis pathway: tyrosinase (TYR), tyrosinase-related protein 1 (TRP1) and DOPAchrome tautomerase (DCT) or tyrosinase-related protein 2 (TRP2). However, only tyrosinase (TYR) is absolutely necessary for melanogenesis, because it is a key enzyme in the process [16,17].

The biologically active compounds from plants have always provided scientists with new sources of useful drugs against infectious diseases. Many plants of the Moraceae family are used in the treatment of infectious diseases [18,19,20,21,22,23,24,25,26,27,28,29,30,31,32,33,34,35,36,37,38,39,40,41,42,43] and hyperpigmentation by inhibition of the tyrosinase enzyme [26,34,41,42,43,44,45,46,47,48,49,50,51,52,53,54,55,56,57,58,59]. *Streblus taxoides* has been reported to have potential antibacterial and antityrosinase activities [42]. The aim of this investigation was to evaluate the antibacterial and antityrosinase activities of the crude extracts and isolated compounds from *S. taxoides*.

## 2. Results and Discussion

### 2.1. Structure Elucidation of Isolated Compounds

Repeated open column chromatography (SiO_2_, ODS and Sephadex LH-20 resins) of the ethyl acetate and methanol fractions from the *Streblus taxoides* wood resulted in the isolation of three 2-arylbenzofuran compounds, including one new compound, named ω-Hydroxymoracin C (**1**), two known arylbenzofuran compounds (**2**–**3**) and two known stilbene compounds (**4**–**5**) (Figure 1). The chemical structures of the isolated compounds were elucidated based on the 1D-NMR and 2D-NMR, MS and IR spectroscopic data. The known compounds were finally identified to be moracin M ( **2**), moracin C (**3**), 3, 4, 3′, 5′-tetrahydroxybibenzyl (**4**) and piceatannol (**5**) by comparing the spectroscopic data with those previously reports [41,60,61,62]. ^1^H-NMR and ^13^C-NMR spectra of new compounds are available for the spectroscopic data.


**ω-Hydroxymoracin C**


Compound **1** was obtained as a brown amorphous powder, soluble in methanol. The UV spectrum in methanol showed absorptions at λ_max_ 210, 317 and 329 nm. The IR spectrum showed the absorption bands at 3216, 1620, 1489, 1444, 1351, 1308, 1150, 1002 and 825 cm^−1^. The EIMS showed a molecular ion peak at m/z 362 corresponding to C_19_H_18_O_5_. The ^1^H-NMR spectrum exhibited five signals of 2-arylbenzofuran and 4 signals of ω-hydroxy prenyl moiety. Three olefinic proton signals of typical *ortho-* and *meta*-coupled patterns of ring A at δ_H_ 7.32 (1H, d, *J* = 8.3 Hz, H-4), 6.72 (1H, dd, *J* = 8.3, 2.2 Hz, H-5) and 6.87 (1H, d, *J* = 2.2 Hz, H-7) with one olefinic proton signal of ring C at δ_H_ 6.82 (1H, d, 0.9, H-3) were identified as a 2-substituted-6-hydroxybenzofuran. One olefinic proton signal at δ_H_ 6.76 (2H, s, H-2′, H-6′) indicated the characteristics of a symmetrical 4-substituted-3,5-dihydroxyphenyl. The ω-hydroxy prenyl moiety proton signals consisted of; one olefinic proton at δ_H_ 5.55 (1H, m, H-2″), two methylenes protons at δ_H_ 3.90 (2H, s, H-5″) and 3.38 (2H, brd, *J* = 7.32 Hz, H-1″) and one methyl proton at δ_H_ 1.81 (3H, s, H-4″). The abovementioned evidence suggests that Compound **1** is a 2-arylbenzofuran with a ω-hydroxy prenyl moiety. The ^13^C-NMR spectrum showed 17 carbon signals. The 2-arylbenzofuran moiety showed: four oxygenated olefinic quaternary signals at δ_C_ 157.61 (C-3′, 5′), 157.17 (C-6) a 156.64 (C-2) and 156.46 (C-7a), five olefinic methine signals at δ_C_ 121.77 (C-4), 113.12 (C-5), 103.81 (C-2′, 6′), 101.33 (C-3) and 98.45 (C-7) and three methylenes signals at δ_C_ 130.45 (C-1′), 123.19 (C-3a) and 116.26 (C-4′). In addition, the following signals of a ω-hydroxy prenyl moiety, confirmed by a previous report [63], were observed: one olefinic quaternary signals at δ_C_ 135.07 (C-3″), one olefinic methines at δ_C_ 125.89 (C-2″), one oxygenated methylenes signal at δ_C_ 69.32 (C-5″), one methylenes signal at δ_C_ 22.95 (C-1″) and one methyls signal at δ_C_ 13.83 (C-4″). The 2-arylbenzofuran structure and the location of the ω-hydroxy prenyl moiety were proven by HMBC-NMR experiments, which suggested all positions and the substitution of the ω-hydroxy prenyl moiety could be formed at the C-4′ of ring B, which was confirmed by the key correlations in the HMBC spectrum (Figure 2).

### 2.2. Antimicrobial Activity

The isolated compounds from *S. taxoides* showed antimicrobial activity against *S. epidermidis*, *S. aureus*, MRSA and *C. acnes.* The MIC and MBC values of the isolated compounds are presented in Table 1. Moracin M showed a weak inhibitory effect against *S. epidermidis*, *S. aureus* and MRSA. Conversely, moracin M derivative such as ω-hydroxymoracin C and moracin C showed a stronger inhibitory effect against *S. epidermidis*, *S. aureus* and MRSA. Moracin M is the arylbenzofuran, while moracin C and ω-hydroxymoracin C are the moracin M derivatives which connect with prenyl and hydroxyprenyl, respectively. The non-prenylated arylbenzofurans exhibited weaker antimicrobial activity than prenylated arylbenzofurans, because the prenyl group could increase the antimicrobial activity of arylbenzofurans [64]. Moreover, moracin C exhibited the effect against *S. aureus* by protein biosynthesis inhibition [63].

### 2.3. Enzymatic Antityrosinase Activity

The isolated compounds were determined on antityrosinase activity by Dopachrom method. 3, 4, 3′, 5′-tetrahydroxybibenzyl (**4**) showed the highest activity against tyrosinase enzyme with a IC_50_ value of 35.65 µg/mL, followed by moracin M (**2**) with a IC_50_ value of 47.34 µg/mL. Conversely, ω-hydroxymoracin C (**1**), moracin C (**3**) and piceatannol (**5**) showed a moderate tyrosinase activity, with IC_50_ values of 109.64, 128.67 and 149.73 µg/mL, respectively. The results are shown in Table 2.

The stilbene group showed an inhibitory effect against tyrosinase enzyme, because it was substituted with the polyhydroxy group, especially at C-2, C-4, C-3′ and C-5′, and a 4-substituted resorcinol structure is important for the tyrosinase inhibitory activity of several stilbenes [65]. Oxyresveratrol has a 4-substituted resorcinol structure and is a substituent with a hydroxy group at C-2, C-4, C-3′ and C-5′, while piceatannol is substituted with a hydroxy group at C-3, C-4, C-3′ and C-5′. Thus, oxyresveratrol showed an antityrosinase activity higher than piceatannol [66]. Moreover, the bibenzyl structure, 2, 4, 3′, 5′-tetrahydroxybibenzyl, which can be obtained from oxyresveratrol through a single-step reduction reaction, showed an inhibitory effect against tyrosinase activity that was higher than oxyresveratrol [47]; similarly, 3, 4, 3′, 5′-tetrahydroxybibenzyl showed an activity higher than piceatannol. Conversely, ω-hydroxymoracin C, moracin M and moracin C are the stilbene derivatives (2-arylbenzofuran), which showed direct activity against the tyrosinase enzyme. The hydroxyl or methoxy group at the C-6 position might mediate the inhibitory activity compared with other 2-arylbenzofurans, i.e., moracin B, moracin J, moracin N and moracin VN [67,68,69,70]. Moreover, the hydroxyl group at the C-3′ and C-5′ position might be important for the tyrosinase inhibitory activity of several 2-arylbenzofuran compared with moracin D, of which the isoprenyl group forms a six-membered ring with a hydroxyl group at C-3′. Thus, moracin D did not display antityrosinase activity [68]. Conversely, the presence or absence of substituent at C-4′ did not affect the tyrosinase inhibitory activity of 2-arylbenzofuran group, so ω-hydroxymoracin C and moracin C which substituted by hydroprenyl and prenyl group, respectively could compare with moracin VN which substituted by dihydroxymethylbutyl [69].

### 2.4. Cell Viability

From the results of the enzymatic investigation, the isolated compounds from *S. taxoides* wood showed antityrosinase activity. Thus, the investigation was extended to cellular experiments. The cell viability was measured first. The results indicated that all sample extracts were not considerable cytotoxic in B16-F1 melanoma cells. Cell viability was still more than 80% at the highest concentration, i.e., 50 μg/mL. The results are shown in Figure 3.

### 2.5. Intracellular Antityrosinase Activity and Melanin Content

The isolated compounds at concentration as 50 µg/mL demonstrated intracellular antityrosinase activity and influenced the melanin content on B16-F1 melanoma cells. The results showed that ω-hydroxymoracin C (**1**), moracin M (**2**) and moracin C (**3**) exhibited antityrosinase activity (Figure 4A). Moreover, the melanin content showed an inverse relationship with antityrosinase activity; an increase of antityrosinase activity could decrease the amount of melanin content (Figure 4B).

The compounds which exhibited enzymatic tyrosinase inhibitory activity showed two different mechanisms in B16-F1 cells. ω-Hydroxymoracin C, moracin M and moracin C showed potential activity with enzymetic antityrosinase and could decrease the melanin content. Conversely, 3, 4, 3′, 5′-tetrahydroxybibenzyl and piceatannol showed potential activity with enzymetic antityrosinase, but increased the melanin content. This might be because ω-hydroxymoracin C, moracin M and moracin C could inhibit the expression and activation of pigment-related proteins, such as the microphthalmia-associated transcription factor [15,17,71,72,73].

### 2.6. Western Blot

The isolated compounds from *S. taxoides* which showed a potential effect of enzymatic antityrosinase activity and were of sufficient quantity to be used for testing were selected to study the melanogenic protein expression in B16-F1 cells.

The results of ω-hydroxymoracin C, moracin M, moracin C, 3, 4, 3′, 5′-tetrahydroxybibenzyl and piceatannol from *S. taxoides* are shown in Figure 5.

The inhibition of melanogenic proteins related to the Wnt-β-catenin-signaling pathway, phosphatidylinositol 3-kinase-Protein Kinase B or PI3K-Akt signaling pathway, cAMP/PKA signaling pathway and mitogen-activated protein kinases or MAPK signaling pathway could decrease melanogenesis. However, the expression of melanogenic proteins of 2-arylbenzofuran group was not clear. The results from the Western blot analysis revealed that moracin M, ω-hydroxymoracin C and moracin C seemed to decrease the melanin content by decreasing microphthalmia-associated transcription factor (MITF), tyrosinase (TYR), tyrosinase-related protein 1 (TRP1) and tyrosinase-related protein 2 (TRP2). Conversely, piceatannol and 3, 4, 3′, 5′-tetrahydroxybibenzyl, a stilbene compound, increased melanin production by increasing microphthalmia-associated transcription factor (MITF) and tyrosinase (TYR), as confirmed with previous reports, since piceatannol exerted its stimulatory effect on melanogenesis by MAP kinase activation and MITF induction of tyrosinase [17,74].

### 2.7. Molecular Docking Experiment

The authors performed this in silico study to predict theoretical molecular interactions between obtained metabolites from *S. taxoides* and tyrosinase, as found in an in vitro enzyme binding assay. In this computational experiment, the authors aim to investigate a possible molecular mode of inhibition. As presented in Figure 6A,B, the authors proposed two different enzyme binding modes from the obtained metabolites found in *S. taxoides*. One was a specific binding mode for moracin derivatives (**1** to **3**), while another was a unique docking site specific to the bibenzyl metabolites group (**4** and **5**), Figure 6B, yellow dot circle.

Integrating visuals from Figure 6A–D with Table 3 showed that the bibenzyl metabolites group entirely docked into an active site of the mushroom tyrosinase (Figure 6A,B, highlighted red area). In detail, these two metabolites also interacted with His263 (one of the histidines holding copper ions) and Ala286 (an amino acid inside the active site). Therefore, the result theoretically hinted at a bibenzyl metabolites’ mode of inhibition as a competitive inhibitor.

On the other hand, in Figure 6A,B,E–G and Table 3, all three moracins derivatives interacted with Ser282, an active residual in a catalytic domain. Furthermore, ω-hydroxymoracin C and moracin C formed chemical bonds with His85 and His259. These two histidines are members of six residues holding copper ions, contributing a catalytic mechanism during enzymatic reaction. Additionally, moracins C and M were chemically bound with neighboring amino acids such as Phe264 and Pro277. This information indicated a possible inhibition mode of predominantly competitive behavior. Interestingly, the authors found that Val283, an amino acid residue at the entrance of the active site, was conserved among all test compounds (Table 3).

Notably, the authors’ molecular docking result was in line with previous experimental reports of moracin and bibenzyl derivatives’ mode on inhibition against mushroom tyrosinase [41,75]. Therefore, the authors’ theoretical simulation through molecular docking proved reliable. 

Furthermore, the three-dimensional chemical structure (Figure 6A,B) showed an outstanding alignment of moracin M (**2**, green color) compared to moracin C (**3**, purple color) and ω-hydroxymoracin C (**1**, the new moracin derivative found in this study, blue color). The difference in docked molecular alignment between moracin derivatives agreed with an earlier in vitro antimushroom tyrosinase activity, showing that moracin M was the most potent inhibitor above moracin C and ω-hydroxymoracin C. Additionally, dimethylallyl moiety, found in both moracin C (**3**) and ω-hydroxymoracin C (**1**), shifted the structural alignment way from moracin M (**2**), the most potent inhibitor, presented in Figure 6B (black arrow). Furthermore, an extra hydroxy group attached to the dimethylallyl moiety of ω-hydroxymoracin C (new derivative) did not impact the molecular alignment based on the authors’ simulation here. It was incorporated with an interaction diagram (Figure 6E,G). No extra hydrogen bond was found in an interaction between ω-hydroxymoracin C (**1**) and amino acid residues, similar to moracin C (**3**). 

Later, the authors rescored the estimated binding energy through Autodock 4 for a more comprehensive energy analysis, as shown in Table 4. The authors found two critical issues after rescoring the energy procedure. First, the estimated docking energy of moracin M (**2**) did not represent the experimental enzyme-binding data. Therefore, the authors did not analyze the obtained energy or compare moracin M’s energy to the other compounds’ docking energy. Second, a default estimated binding energy obtained from Autodock 4 was poorly correlated (R^2^ = 0.46) with the experimental enzyme-binding data presented earlier. Therefore, the authors manually modified the estimated binding energy obtained from the program by ignoring torsion-free energy. The authors ignored the torsion-free energy function, because it did not agree with the experimental data the most. As a result, ignoring a torsion-free function, a correlation coefficient was improved from R^2^ of 0.46 to R^2^ of 0.89. Therefore, the author used a modified binding energy instead of a default one obtained from the program for our analysis.

Based on Table 4, the total intermolecular interaction energies between ω-hydroxymoracin C (**1**) and moracin C (**3**) were similar. However, the significant energy function contributing to a more favorable binding energy (without the torsion-free function mentioned earlier) from ω-hydroxymoracin C (**1**) was a lower total internal energy. This function describes an advantage of the internal flexibility of a small molecule with rotatable bonds. Referring to an interaction diagram earlier, since no extra intermolecular interaction was found, an internal factor was a rational explanation for a superior inhibitory effect of ω-hydroxymoracin C over moracin C (**2**). 

On the other hand, in bibenzyl derivatives, the total internal energies between 3,3′,4,5′-tetrahydroxybibenzyl **(4**) and piceatannol (**5**) were similar. This indicated that an intermolecular factor contributed to a more favorable binding affinity of 3,3′,4,5′-tetrahydroxybibenzyl **(4**). The authors found that a van der Waal and hydrogen bond (vdW+Hbond) function was the significant factor. Additionally, the vdW+Hbond energy function corresponded to interaction diagrams (Figure 6C,D). Based on the diagram, 3,3′,4,5′-tetrahydroxybibenzyl **(4**) formed six intermolecular bonds with amino acids in the active site. Four bonds, i.e., pi-alkyl, pi-pi stacked, pi-pi T-shaped and pi-sigma, were similar to piceatannol (**5**). Two extra bonds were unique and only presented in the 3,3′,4,5′-tetrahydroxybibenzyl interaction. One was pi-alkyl (pink dash line in Figure 6C) and another was a hydrogen bond (green dash line in Figure 6C). Therefore, these two outstanding bonds provided an explanation of why 3,3′,4,5′-tetrahydroxybibenzyl (**4**) was more favorably bound with tyrosinase than piceatannol (**5**).

In conclusion, the authors’ simulation experiment provided a theoretical explanation supporting an in vitro enzyme-binding experimental outcome found earlier. To sum up, the simulation showed that structural variation affected enzyme-inhibitor binding structurally and energetically, either internally or externally.

## 3. Materials and Methods

### 3.1. Plant Materials

The wood of *Streblus taxoides* (Heyne ex Roth) Kurz was collected from Rajjaprabha Dam, Surat Thani Province, Thailand. It was identified by a botanist of the Southern Literature Botanical Garden, and the voucher specimen number was SKP 117 19 20 01. The sample specimen was deposited at the Department of Pharmacognosy and Pharmaceutical Botany, Faculty of Pharmaceutical Sciences, Prince of Songkla University, Thailand.

### 3.2. Extraction and Isolation

In total, 18 kg of dried powder from *S. taxoides* wood was macerated repeatedly with petroleum ether for 3 days, which was repeated three times. The filtrated sample was evaporated by rotary evaporator under reduced pressure at below 40 °C to yield a petroleum ether extract. Then, the marc was macerated with ethyl acetate and methanol for 3 days, which was repeated three times each, and boiled with H_2_O, respectively. Removal of organic solvents gave an ethyl acetate extract, methanol extract and H_2_O extract, respectively.

From the screening result [42], the ethyl acetate and methanol crude extracts showed activity against tyrosinase enzyme and microbe. Thus, these crude extracts were selected for further phytochemical investigation by using chromatographic techniques. Initially, 22 g of ethyl acetate extract was isolated by quick column chromatography, while the gradient of dichloromethane, ethyl acetate and methanol were used as an eluent. The interesting fractions were E, H, J and K. Moreover, fraction C, D, E and F were the interesting fractions which were fractionated by quick column chromatography from 50 g of methanol extract, using the gradient of hexane, ethyl acetate and formic acid as an eluent. All interesting fractions were selected to further isolation and purification. However, many isolated compounds from interesting fractions did not stable, they were easy to degrade. The steps of isolation are summarized in Figure 1.

### 3.3. Spectroscopic Data

IR spectra were obtained from a Perkin Elmer FT-IR Spectrum One spectrometer, using Potassium bromide disc to determine the spectra.

Electron Impact Mass Spectra (EIMS) were measured on a Thermo Finnigan MAT 95 XL mass spectrometer.

^1^H and ^13^C spectra were obtained with a Fourier Transform NMR Spectrometer (1H-NMR 500 MHz and 13C-NMR 125 MHz), model UNITY INNOVA Varian.

(1)ω-Hydroxymoracin C; C_19_H_18_O_5_

The IR spectrum showed the absorption bands at 3216, 1620, 1489, 1444, 1351, 1308, 1150, 1002 and 825 cm^−1^ and gave a molecular ion at 326 *m/z* in the EIMS. ^1^H-NMR (500 MHz, CD_3_OD, δ_H_); 7.32 (1H, d, *J* = 8.3 Hz, H-4), 6.87 (1H, d, *J* = 2.2 Hz, H-7), 6.82 (1H, d, *J* = 0.9 Hz, H-3), 6.78 (2H, s, H-2′, H-6′), 6.72 (1H, dd, *J* = 8.3, 2.2 Hz, H-5), 5.55 (1H, m, H-2″), 3.90 (2H, s, H-5″), 3.38 (2H, brd, *J* = 7.32 Hz, H-1″), 1.81 (3H, s, H-4″); ^13^C-NMR (125 MHz, CD_3_OD, δ_C_); 157.61 (C-3′, 5′), 157.17 (C-6), 156.64 (C-7a), 156.46 (C-2), 135.07 (C-3″), 130.45 (C-1′), 125.89 (C-2″), 123.19 (C-3a), 121.77 (C-4), 116.27 (C-4′), 113.12 (C-5), 103.81 (C-2′, 6′), 101.33 (C-3), 98.45 (C-7), 69.32 (C-5″), 22.95 (C-1″), 13.83 (C-4″). 

(2)Moracin M; C_14_H_10_O_4_ [41]

The IR spectrum showed the absorption bands at 3246, 2925, 1613, 1489, 1292 and 1149 cm^−1^. ^1^H-NMR (500 MHz, DMSO, δ_H_); 7.38 (1H, d, *J* = 8.5 Hz, H-4), 7.06 (1H, d, *J* = 0.98 Hz, H-3), 6.91 (1H, dd, *J* = 2.2, 0.97 Hz, H-7), 6.72 (1H, dd, *J* = 8.5, 2.2 Hz, H-5), 6.67 (2H, d, *J* = 1.9 Hz, H-2′, H-6′), 6.20 (1H, t, *J* = 2.2 Hz, H-4′); ^13^C-NMR (125 MHz, CD_3_OD, δ_C_); 158.94 (C-3′, 5′), 155.87 (C-6), 155.42 (C-7a), 154.12 (C-2), 131.82 (C-1′), 121.25 (C-4), 120.94 (C-3a), 112.61 (C-5), 102.80 (C-4′), 102.47 (C-2′, C-6′), 101.69 (C-3), 97.62 (C-7). 

(3)Moracin C; C_19_H_18_O_4_ [60]

The IR spectrum showed the absorption bands at 3216, 2693, 1606, 1489, 1357 and 1150 cm^−1^. ^1^H-NMR (500 MHz, DMSO, δ_H_); 9.51 (1H, s, 4-OH), 9.29 (2H, s, 3′-OH, 5′-OH), 7.37 (1H, d, *J* = 8.2 Hz, H-4), 6.90 (1H, brd, *J* = 1.9 Hz, H-7), 6.89 (1H, d, *J* = 0.7 Hz, H-3), 6.73 (2H, s, H-2′, H-6′), 6.71 (1H, dd, *J* = 8.3, 1.9 Hz, H-5), 5.17 (1H, m, H-2″), 3.19 (2H, brd, *J* = 6.8 Hz, H-1″), 1.70 (3H, brs, H-4″), 1.60 (3H, brs, H-5″); ^13^C-NMR (125 MHz, CD_3_OD, δ_C_); 156.39 (C-3′, C-5′), 155.69 (C-6), 155.30 (C-7a), 154.34 (C-2), 129.75 (C-3″), 128.14 (C-1′), 123.31 (C-2″), 121.09 (C-3a), 120.99 (C-4), 115.09 (C-4′), 112.48 (C-5), 102.43 (C-2′, C-6′), 100.65 (C-3), 97.55 (C-7), 25.64 (C-4″), 22.21 (C-1″), 17.84 (C-5″).

(4)3, 4, 3′, 5′-Tetrahydroxybibenzyl; C_14_H_14_O_4_ [61]

The IR spectrum showed the absorption bands at 3435, 1616, 1521, 1468, 1285 and 1160 cm^−1^. ^1^H-NMR (500 MHz, DMSO, δ_H_); 6.64 (1H, d, *J* = 8.0, H-5), 6.60 (1H, d, *J* = 1.9, H-2), 6.48 (1H, dd, *J* = 8.0, 2.2, H-6), 6.12 (2H, d, *J* = 2.4, H-2′, H-6′), 6.07 (1H, t, *J* = 2.2, H-4′), 2.67 (1H, 2, m, H-α′), 2.66 (1H, 2, m, H-α); ^13^C-NMR (125 MHz, CD_3_OD, δ_C_); 159.29 (C-3′, C-5′), 146.01 (C-3), 145.70 (C-4), 144.26 (C-1′), 135.00 (C-1), 120.70 (C-6), 116.24 (C-5), 108.04 (C-2′, C-6′), 101.15 (C-4′), 39.51 (C-α′), 38.28 (C-α).

(5)Piceatannol; C_14_H_12_O_4_ [62]

The IR spectrum showed the absorption bands at 3368, 1600, 1520, 1444, 1285 and 1160 cm^−1^. ^1^H-NMR (500 MHz, DMSO, δ_H_); 6.96 (1H, d, *J* = 1.9 Hz, H-2), 6.87 (1H, d, *J* = 16.1 Hz, H-α′), 6.82 (1H, dd, *J* = 8.3, 1.9 Hz, H-6), 6.73 (1H, d, *J* = 16.1 Hz, H-α), 6.72 (1H, d, *J* = 8.3 Hz, H-5), 6.42 (2H, d, *J* = 2.2 Hz, H-2′, H-6′), 6.15 (1H, t, *J* = 2.2 Hz, H-4′).

### 3.4. Antimicrobial Activity Assay

Microorganisms that cause skin infection, such as *Staphylococcus aureus* (ATTC 25923), *Staphylococcus epidermidis* (TISTR 517), *Cutibacterium acnes* (DMST 14916) and methicillin-resistant *Staphylococcus aureus* (DMST20654), were selected for this study. The isolated compounds were determined for minimum inhibitory concentration (MIC) in a 96-well plate by modified broth microdilution method and minimum bactericidal concentration (MBC) [76,77]. Oxacillin was used as positive controls for *S. aureus, S. epidermidis* and *C. acnes*, while vancomycin was used for MRSA.

### 3.5. Enzymetic Antityrosinase Activity Assay

The tyrosinase activity was measured by the Dopachrom method [44,78] using L-DOPA as a substrate. Briefly, 140 μL phosphate buffer (pH 6.8), 20 μL sample solution, and 20 μL tyrosinase solution (203.3 unit/mL) were mixed at 25 °C for 10 min, after which 20 μL of 0.85 mM L-Dopa was added. The optical density (OD) was measured at 492 nm. After incubation at 25 °C for 20 min, the optical density was measured again. The percentage of tyrosinase inhibition was calculated with Equation (1):Tyrosinase inhibition (%) = (1 − [OD_492_ of sample/OD_492_ of control]) × 100(1)

Kojic acid and water extract of *Artocarpus lacucha* wood were used as positive controls and dimethyl sulfoxide (DMSO) was used as a negative control.

### 3.6. Cell Culture

The murine B16-F1 melanoma cells (CLS-400122, CLS Cell Lines Service GmbH, Germany) were cultured in Dulbecco’s modified Eagle’s medium, which was supplemented with 10% fetal bovine serum in a humidified incubator at 37 C with 5% CO_2_. When cells reached 70–80% confluence cell viability, cellular tyrosinase activity and melanin content were measured [79,80,81,82].

### 3.7. Cell Viability Assay

Cell viability was determined by sulforhodamine B (SRB) assay. Briefly, the B16-F1 cells were seeded at a density of 5 × 10^3^ cells/well on 96-well plates and cultured for 24 h. Then, the cells were treated with test samples and 0.5% DMSO for negative control. After 48 h of incubation, cells were fixed with 10% trichloroacetic acid (TCA) and incubated at 4 °C, for 1 h. After that, cells were strained with 0.45% SRB. Then, 10 mM Tris base was added on strained cells, after which SRB color was dissolved by shaking. Optical densities were determined at 492 nm. The percentage of cell viability was calculated.

### 3.8. Intracellular Antityrosinase Activity and Melanin Content Assays

The B16-F1 cells were seeded at a density of 3 × 10^5^ cells/well on 12-well plates and cultured for 12 h. Cells were treated with test samples and control (0.5% DMSO). After 48 h of incubation, cells were lysed with RIPA and centrifuged at 14,000 rpm for 20 min (4 °C) to separate the cell pellet and supernatant. The supernatant was collected and the protein content was determined by the Bradford method using bovine serum albumin as standard [83]. The supernatant was incubated and 2 mg/mL L-Dopa was added to a 96-well plate at 25 °C for 1 h. After that, optical densities were measured at 492 nm. Then, tyrosinase inhibition was calculated, while the cell pellet was dissolved with 1 M NaOH and incubated at 55 °C for 1 h. Melanin concentration was calculated by comparing the absorbance at 475 nm using a standard curve of synthetic melanin.

### 3.9. Western Blot Analysis

The protein content of the supernatant was quantified by the Bradford method using bovine serum albumin as standard. Equal amounts of protein were separated by 40% acrylamide/Bissolution (InvitrogenThermoFisher Scientific, Waltham, MA, USA) and transferred onto nitrocellulose membranes (BIO-RAD Laboratories, Feldkirchen, Germany). The membranes were probed with antibodies against microphthalmia-associated transcription factor; MITF (ThermoFisher Scientific #MA5-16214, Waltham, MA, USA), tyrosinase (ThermoFisher Scientific #35-6000, Waltham, MA, USA), tyrosinase-related protein 1; TRP1 (ThermoFisher Scientific #OSR00085W, Waltham, MA, USA) and tyrosinase-related protein 2; TRP2 (ThermoFisher Scientific #PA5-36485, Waltham, MA, USA). The proteins were detected using an enhanced chemiluminescence kit (BIO-RAD Laboratories, Hercules, CA, USA). Quantitative analysis was performed using a digital imager (UPV UVP, VisionWork^T^ LS, Image Acquisition & Analysis Software). The method applied was modified from Western Blot analysis for UGT1A family by the co-author, Asst. Prof. Dr. Wandee Udomuksorn, Pharmacology Program, Division of Health and Applied Science, Faculty of Science, Prince of Songkla University, Thailand., Department of Clinical Pharmacology, Flinders Medical Center.

### 3.10. Molecular Docking Experiment

The authors downloaded nearly all compounds’ structures from the PubChem database (https://pubchem.ncbi.nlm.nih.gov/, accessed on 15 December 2022), except for ω-hydroxymoracin C (a new derivative). Therefore, the authors created ω-hydroxymoracin C from moracin C (Compound **2**, Pubchem CID 155248) using the Avogadro program, version 1.2.0. [84]. In addition, the authors provided all PubChem CIDs of all obtained compound structures in a Appendix A (Appendix A). Finally, before the docking experiment, all compounds were geometrical and force field (MMFF94s) optimizations through the same Avogadro program used earlier, following the authors’ previous publications [85,86,87].

On the other hand, the authors obtained the mushroom tyrosinase crystal structure, PDB ID: 2y9x, from the Protein Databank or PDB (https://www.rcsb.org/, accessed on 15 December 2022) [88]. After that, the authors used Autodock Tools version 1.5.6 to prepare tyrosinase properly for the docking experiment [89]. Next, the authors extracted a native ligand (tropolone) that came with the tyrosinase crystal structure. Later, the authors used it to navigate a catalytic pocket and validate an established docking protocol via a re-docking approach. Only the docking protocol provided a root-mean-square deviation (RMSD) value less than 2 Å was used after redocking tropolone back to its original position. Finally, the authors provided the validated docking protocol that passed the criterion in a Appendix A (Appendix A).

The authors used Autodock Vina version 1.1.2 to perform the docking experiment in this study [90]. All docking parameters were set as a default value. However, some parameters were changed, such as the exhaustiveness value adjusted up to twenty-four and the corrected numbers of docking pose sets to twenty. Finally, the authors designed a docking grid box as x = −10.1, y = −28.7 and z = −43.4 with a size of 18 Å × 18 Å × 18 Å.

For post-docking analysis, the authors used the Chimera program version 1.11.2 for ligand-protein three-dimensional visualization [91] and applied Discovery Studio free version 20.1.0.19295 for the intermolecular interactions diagram [92].

## 4. Conclusions

The two new compounds 2-arylbenzofuran and ω-hydroxymoracin C and four known compounds, i.e., moracin M, moracin C, 3, 4, 3′, 5′-tetrahydroxybibenzyl and piceatannol, were isolated from the *S. taxoides* wood. All isolated compounds showed potential activity with enzymatic antityrosinase, consistent with the data from molecular docking, indicating a possible inhibition mode of predominantly competitive behavior. 

The isolated compounds which exhibited enzymatic tyrosinase inhibitory activity consisted of two different mechanisms in B16-F1 cell as Western blot confirmation. ω-Hydroxymoracin C, moracin M and moracin C showed potential activity with enzymatic antityrosinase and could decrease melanin content by downregulated the melanogenic protein expression. Conversely, 3, 4, 3′, 5′-tetrahydroxybibenzyl and piceatannol showed potential activity with enzymatic antityrosinase but increase melanin content by upregulating the melanogenic protein expression.

Moreover, ω-hydroxymoracin C and moracin C exhibited potential effects on antimicrobial activity. This is the first report of the phytochemical composition of *S. taxoides*, with new compounds and their bioactivities. The results indicated the high potential of some isolated compounds, which might be utilized as the new alternative lead compounds for further research of whitening and/or antiacne agents.

## Data Availability

Data sharing is not applicable.

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
