# Peer review of "Chemical Constituents from Streblus taxoides Wood with Their Antibacterial and Antityrosinase Activities Plus in Silico Study"

_antibiotics, 2023, doi:10.3390/antibiotics12020319_

Round 1

Reviewer 1 Report

The manuscript needs extensive English editing, at many times it is incomprehensible. There is irrelevant and redundant information too. Figure 1 is not needed.  Known compounds activity dominates. 

Author Response

Reply to Reviewer 1

Comments and Suggestions for Authors

The manuscript needs extensive English editing, at many times it is incomprehensible. There is irrelevant and redundant information too. Figure 1 is not needed.  Known compounds activity dominates. 

### Thank you for your suggestion. The faculty’s English facilitator helped us to edit this revised manuscript. Figure 1 was deleted.

Reviewer 2 Report

In order to be accepted for publication in antibiotics journal, the article must comply with the following corrections described below.

The abstract must be rewritten, since it presents inconsistencies in different sentences, for example the sentence of lines 16-18 must be rewritten.

The title of the article is very broad, I suggest that it be more specific. For example, include in it the biological activities that were explored.

In the introduction, I suggest including a paragraph that better describes the relationship between antimicrobial and antirosinase activity, discuss at least 5 papers

You should include the NMR-H and 13C spectra and high resolution mass spectra in the supplementary information section. In the case of compounds that are not new, please include in the experimental section the bibliographical reference where the signals of the particular compound have been described.

Author Response

Reply to Reviewer 2

Comments and Suggestions for Authors

In order to be accepted for publication in antibiotics journal, the article must comply with the following corrections described below.

The abstract must be rewritten, since it presents inconsistencies in different sentences, for example the sentence of lines 16-18 must be rewritten.

### Thank you very much. The abstract was rewritten.

The title of the article is very broad, I suggest that it be more specific. For example, include in it the biological activities that were explored.

### Thank you very much. The title was edited. The new title is “Chemical constituents from Streblus taxoides wood with their antibacterial and antityrosinase activities plus in silico study

In the introduction, I suggest including a paragraph that better describes the relationship between antimicrobial and antirosinase activity, discuss at least 5 papers

### I edited it by described the relationship between antimicrobial, inflammation and antityrosinase activity.

You should include the NMR-H and 13C spectra and high resolution mass spectra in the supplementary information section. In the case of compounds that are not new, please include in the experimental section the bibliographical reference where the signals of the particular compound have been described.

### I added 1H, 13C and 2D-NMR spectra in the supplementary information section.

### All known compound data with their references were written in the experimental section (3.3 Spectroscopic Data)

Reviewer 3 Report

The article entitled "Chemical constituents from Streblus taxoides wood and their bioactivities within silico study" is written as per journal style. a few corrections like, grammatical correction and sentences framing are required. 

other correction like

line 40-41 " Some mediators of inflammation, such as prostaglandin E2 40 (PGE2) it can be stimulate the melanogenesis to produce melanin pigment" revise it

line 180-182-->2.4. Cell Viability: Provide statistical analysis in the figure.

Material methods : Mention the name of equipment and operating frequency like 400MHz, 100 MHz

source of "B16-F1 melanoma cell lines"

Figure 7: Discuss the molecular interactions with different aminoacids and the energy required.

Author Response

Reply to Reviewer 3

The article entitled "Chemical constituents from Streblus taxoides wood and their bioactivities with in silico study" is written as per journal style. a few corrections like, grammatical correction and sentences framing are required.

other correction like

line 40-41 "Some mediators of inflammation, such as prostaglandin E2 40 (PGE2) it can be stimulate the melanogenesis to produce melanin pigment" revise it

### I described more the relationship between antimicrobial, inflammation and antityrosinase activity

line 180-182-->2.4. Cell Viability: Provide statistical analysis in the figure.

### It was edited.

Material methods: Mention the name of equipment and operating frequency like 400MHz, 100 MHz

### I had added all data.

source of "B16-F1 melanoma cell lines"

### I had added all data.

Figure 7: Discuss the molecular interactions with different aminoacids and the energy required.

### It has done.

Reviewer 4 Report

Overall Comment on the review:

In the Article entitled Chemical constituents from Streblus taxoides wood and their bioactivities with in silico study authors presented a comprehensive study of the isolation of the pure compounds, one of them representing a new natural product, and a broad spectrum of biological activity of five isolated natural products. The strongest point of the manuscript is that the presented data gives us a picture of the therapeutic potential of the S. taxoides that is already previously noted by Dej-adisai and co-workers (Pharmacognosy Magazine, 15 (65), 2019), offering insights for future research. The additional strong point is in silico docking experiments that predicted theoretical molecular interactions between obtained metabolites and tyrosinase. Overall, I enjoyed reading this article. The weakest point is that extensive editing of the English language is required.

Having said that, in my opinion, this manuscript can be recommended for publication in such a reputable scientific journal after major revisions that include:

1) Extensive edition of the English language.

2) in silico in the title and throughout the manuscript SHOULD BE in italic (in silico).

3) (Compound 1, etc) SHOULD BE (1), etc. COMMENT It is not necessary to write the word compound. Please change that in all places throughout the manuscript.

4) You mentioned that the chemical structures of the isolated compounds were elucidated based on the 1D-NMR and 2D-NMR, MS, and IR spectroscopic data. COMMENT Please add crude 1D-NMR, MS, and IR spectra of the isolated compounds in the Supplementary data. For the new compound (ω-hydroxymoracin C) please add also 2D-NMR spectra.

5) In the manuscript you write the capital letter for the isolated compounds. COMMENT please write a small letter for the compounds, except when the name of the compound is at the beginning of the sentence.

6) Even detailed NMR spectral analysis of the new compound is presented, in my opinion, authors miss the opportunity to analyze the conformation of the double bond in the hydroxylated prenyl moiety. Is it E or Z? Conformation of the double bond can have a great impact on biological activity and also is important for docking experiments.

7) In the Materials and methods please add a part with the conditions of the NMR and MS analysis.

8) In the part Spectral data, besides NMR data, please add data from the MS and IR analysis.

9) Please avoid the phrase: The authors used … etc.

Author Response

Reply to Reviewer 4

Comments and Suggestions for Authors

Overall Comment on the review:

In the Article entitled Chemical constituents from Streblus taxoides wood and their bioactivities with in silico study authors presented a comprehensive study of the isolation of the pure compounds, one of them representing a new natural product, and a broad spectrum of biological activity of five isolated natural products. The strongest point of the manuscript is that the presented data gives us a picture of the therapeutic potential of the S. taxoides that is already previously noted by Dej-adisai and co-workers (Pharmacognosy Magazine, 15 (65), 2019), offering insights for future research. The additional strong point is in silico docking experiments that predicted theoretical molecular interactions between obtained metabolites and tyrosinase. Overall, I enjoyed reading this article. The weakest point is that extensive editing of the English language is required.

Having said that, in my opinion, this manuscript can be recommended for publication in such a reputable scientific journal after major revisions that include:

1) Extensive edition of the English language.

### The faculty’s English facilitator helped us to edit this revised manuscript.

2) in silico in the title and throughout the manuscript SHOULD BE in italic (in silico).

### I am sorry, I followed the journal format, is not italic.

3) (Compound 1, etc) SHOULD BE (1), etc. COMMENT It is not necessary to write the word compound. Please change that in all places throughout the manuscript.

### It was edited throughout the revised manuscript.

4) You mentioned that the chemical structures of the isolated compounds were elucidated based on the 1D-NMR and 2D-NMR, MS, and IR spectroscopic data. COMMENT Please add crude 1D-NMR, MS, and IR spectra of the isolated compounds in the Supplementary data. For the new compound (ω-hydroxymoracin C) please add also 2D-NMR spectra.

### I had added all data.

5) In the manuscript you write the capital letter for the isolated compounds. COMMENT please write a small letter for the compounds, except when the name of the compound is at the beginning of the sentence.

### It was edited.

6) Even detailed NMR spectral analysis of the new compound is presented, in my opinion, authors miss the opportunity to analyze the conformation of the double bond in the hydroxylated prenyl moiety. Is it E or Z? Conformation of the double bond can have a great impact on biological activity and also is important for docking experiments.

### Thank you for your suggestion. Unfortunately, this new compound was amorphous powder and we have got trace amount from isolation, it was not enough for further experiment i.e. optical rotation and so on.

7) In the Materials and methods please add a part with the conditions of the NMR and MS analysis.

### I had added all data.

8) In the part Spectral data, besides NMR data, please add data from the MS and IR analysis.

### I had added all data.

9) Please avoid the phrase: The authors used … etc.

### Thank you, I edited it.

Round 2

Reviewer 1 Report

MS is improved now, though scheme 1 is still not resolved well to see.

The HMBC (2D-NMR) correlations (Figure 2) only for few atoms, authors need to show all connectivities.

Moderate improvement in the language is required.

Author Response

Reply to Reviewer 1-RV2

Comments and Suggestions for Authors

MS is improved now, though scheme 1 is still not resolved well to see.

### Thank you for your suggestion. The new scheme 1 was added in the revised MS.

The HMBC (2D-NMR) correlations (Figure 2) only for few atoms, authors need to show all connectivities.

### Thank you for your suggestion. The new Figure 2 was added in the revised MS and showed all correlations.

Moderate improvement in the language is required.

### Thank you for your suggestion. The faculty’s English facilitator helped us to edit this revised MS.

Reviewer 2 Report

Manuscript accepted for publication

Author Response

Reply to Reviewer 2-RV2

Comments and Suggestions for Authors

Manuscript accepted for publication

### Thank you very much.

Reviewer 4 Report

In the revised version of the manuscript (entitled Chemical constituents from Streblus taxoides wood with their antibacterial and antityrosinase activities plus in silico study) the authors changed the manuscript, according to the reviewer’s suggestions, and significantly improved the first version, which I encourage to be accepted in the present form.

Author Response

Reply to Reviewer 4-RV2

Comments and Suggestions for Authors

In the revised version of the manuscript (entitled Chemical constituents from Streblus taxoides wood with their antibacterial and antityrosinase activities plus in silico study) the authors changed the manuscript, according to the reviewer’s suggestions, and significantly improved the first version, which I encourage to be accepted in the present form.

### Thank you very much.
